# Intragenomic sequence variations in the second internal transcribed spacer (ITS2) ribosomal DNA of the malaria vector *Anopheles stephensi*

Shobhna Mishra[1], Gunjan Sharma[1], Manoj K. Das[2], Veena Pande[3], Om P. Singh[1]*

1 National Institute of Malaria Research, New Delhi, India, 2 Field Unit, National Institute of Malaria Research, Itki, Ranchi, India, 3 Department of Biotechnology, Kumaun University, Nainital, Uttarakhand, India

* dr.opsingh@gmail.com, singh@mrcindia.org

## Abstract

Second Internal Transcribed Spacer (ITS2) ribosomal DNA (rDNA) sequence is a widely used molecular marker for species-identification or -delimitation due to observed concerted evolution which is believed to homogenize rDNA copies in an interbreeding population. However, intra-specific differences in ITS2 of *Anopheles stephensi* have been reported. This study reports the presence of intragenomic sequence variation in the ITS2-rDNA of *An. stephensi* and hypothesizes that observed intra-specific differences in this species may have resulted due to ambiguous DNA sequence-chromatogram resulting from intragenomic heterogeneity. *Anopheles stephensi* collected from different parts of India were sequenced for complete ITS2 and the variable region of 28S-rDNA (d1-d3 domains). Intragenomic variations were found in ITS2 region of all *An. stephensi* sequenced, but no such variation was observed in d1 to d3 domains of 28S-rDNA. Cloning and sequencing of ITS2 through the d3 domain of the 28S region of rDNA from representative samples from northern, central, and southern India confirmed the presence of intragenomic variation in ITS2 due to transitions at three loci and two bp indel in a di-nucleotide microsatellite locus. Multiple haplotypes were observed in ITS2 raised from such variations. Due to the absence of detectable intragenomic sequence variation in the d1 to d3 domain of 28S rDNA of *An. stephensi*, this region can serve as an ideal reference sequence for taxonomic and phylogenetic studies. The presence of intragenomic variation in rDNA should be carefully examined before using this as a molecular marker for species delimitation or phylogenetic analyses.

## Introduction

*Anopheles stephensi* is a major malaria vector in India, especially in an urban setting. This vector is predominantly found in urban areas due to their preference to breed in clean water especially in cement tanks, containers, overhead tanks, cisterns, fountain tanks, and building

**Data Availability Statement:** The data underlying this study are available on GenBank (Accession nos: MW676288—MW676295; MW732930—MW732931).

**Funding:** This research was supported by the Department of Science & Technology, India, https://dst.gov.in/ (OPS) and ICMR-Senior Research Fellowship (SM and GS). The funders had no role in study design, data collection and analysis, decision to publish, or preparation of the manuscript.

**Competing interests:** The authors have declared that no competing interests exist.

construction sites [1] which are abundant in urban areas. Perhaps due to its adaptation to such breeding habitats, this species has invaded several neighbouring territories. Because of the invasive nature of *An. stephensi*, coupled with the high infectivity rate to human-*Plasmodium* species [2–4], this malaria vector has now gained global health attention.

Historically, *An. stephensi* was recorded from the Middle-East and South Asia region (Afghanistan, Bahrain, Bangladesh, China, Egypt, India, Iran, Iraq, Oman, Pakistan, Saudi Arabia, Thailand) [1]. According to Sharma [5] *An. stephensi* is an invasive species that appeared first in the port cities in India and invaded the riparian town first and then the towns with excessive wells. During the last few decades, invasions of this species have been reported in Lakshadweep (India) [6], Sri Lanka [7–9], Republic of Djibouti [10], Ethiopia [11, 12] and Sudan [13]. *Anopheles stephensi* is now established in the Horns of Africa and contributing to local malaria transmission [14, 15]. As a consequence of reported invasions, Word Health Organization [13] alerted WHO-member-states and their implementing partners of affected countries to take immediate action.

Molecular markers, mainly the second internal transcribed spacer (ITS2) ribosomal DNA (rDNA) and cytochrome c oxidase, subunit 1 (COI) have been frequently used for the identification and confirmation of invasion of *An. stephensi* in several countries [7, 8, 10–12, 15]. Another nuclear marker, intron of odorant-binding protein 1 (OBP1), has been reported to identify biological variants of *An. stephensi* [16] and based on this marker, the presence of three sibling species in *An. stephensi* has been speculated [17]. Ribosomal DNA is the preferred and most widely used molecular marker for the delimitation of species and developing species-diagnostic assays being highly conserved in the interbreeding population due to the homogenization of copies which is thought to be achieved due to concerted evolution [24]. Ribosomal DNA is comprised of tandemly arranged units of ETS, 18S, ITS1, 5.8S, ITS2, and 28S, in several hundreds of copies in insects. DNA sequences of variable regions of 28S-rDNA, IGS and ITS2 in particular, have been frequently used for species identification, species delimitation, and phylogenetic analyses. 28S rDNA is a relatively conserved region, however, some domains (d2 and d3) are variable and have been used as markers for species differentiation [18–23]. The second internal transcribed spacer region, on the other hand, which is relatively highly variable among different species because of its high evolution rate, is the most extensively used taxonomic marker for species identification and phylogenetic studies. However, in some species, intragenomic variations have been reported with the presence of multiple haplotypes. Such intragenomic variation poses a challenge in DNA sequencing especially when indel is present. The presence of indel leads to ambiguous sequencing result starting from the point of indel onwards which can be resolved through cloning [24]. The presence of such indels has resulted in the submission of incorrect sequences in GenBank or published reports. Such a phenomenon has recently been observed in the case of *An. subpictus* [25] and it was noted that all the reported intra-specific differences in a molecular form (Form A) of *An. subpictus* was in fact due to ambiguous DNA sequence chromatogram resulting from the presence of indel in one of the two haplotypes of ITS2 present in each individual. The selection of the molecular region of rDNA, which can serve as reference sequences for the species identification, is therefore important and a region exhibiting intragenomic heterogeneity should be avoided. Moreover, recording of intragenomic sequence variation in rDNA becomes important while reporting sequence data to avoid ambiguities in taxonomic and phylogenetic studies.

In the case of *An. stephensi*, numerous ITS2 sequences are available in the GenBank and research publications [2, 8, 10, 26–28]. However, polymorphism in length and nucleotide sequence has been observed in sequences reported from Iran [26], India [28], and in several GenBank sequences. Such intraspecific polymorphism is intriguing. In some other reports

from India [27], Saudi Arabia [29], Ethiopia [11] and Sri Lanka [8], however, no intraspecific differences were noted. One plausible reason, which explains reporting of polymorphism, is the presence of intragenomic sequence variation which has not been taken care of during DNA sequence analysis.

In this study, we characterized partial 5.8S, complete ITS2, and partial 28S (d1 to d3 domains) of *An. stephensi* to investigate intragenomic variations and delineate the region of rDNA suitable for molecular taxonomy that can be correctly sequenced without the need for cloning.

## Material and methods

### Mosquito samples

*Anopheles stephensi* mosquitoes were collected from different parts of India, i.e., Gurugram (Haryana), Nuh (Haryana), Alwar (Rajasthan), and New Delhi from northern India, Ranchi (Jharkhand), Raipur (Chattishgarh), and Gadhchiroli (Maharashtra) from central India, Goa, Bangalore (Karnataka) Mangalore (Karnataka), Chennai (Tamil Nadu) and Mysuru (Karnataka) from southern India (Fig 1). The geographical coordinates of collection sites have been provided in Table 1. Individual mosquitoes were preserved in a microfuge tube containing a piece of dehydrated silica gel and transported to the laboratory at Delhi. Mosquitoes were identified using keys by Christophers [30] before DNA isolation.

### DNA isolation and direct sequencing

DNA was isolated from individual female mosquitoes using the method by Livak [31]. A part of rDNA spanning part of 5.8S rDNA, complete ITS2, and part of 28S rDNA (d1 to d3 domains) was amplified by using primers ITS2A (5'-TGT GAA CTG CAG GAC ACA T-3') [32] and D3B (5'-TCG GAA GGA ACC AGT TAC TA-3') [33]. The PCR reaction mixture (20 μL) contained 1X buffer, 1.5 mM MgCl$_2$, 200 μM of each dNTP, 0.25 μM of each primer, 0.50 unit of Taq polymerase (GoTaq®, Promega Corporation Inc), and 0.5 μL of DNA template. The PCR conditions were: an initial denaturation step at 95 ˚C for 3 min, followed by 35 cycles each with a denaturation step at 95 ˚C for 30 sec, annealing step at 55 ˚C for 30 sec and extension step at 72 ˚C for 1.5 min, followed by a final extension at 72 ˚C for 7 min. The quality of PCR product was checked on 1.5% agarose gel under UV gel documentation unit. The PCR products were subjected to DNA sequencing using Sanger's method. The PCR products were treated with Exo-Sap (ExoSAP-IT™, Thermo Fisher, USA) to remove unutilized primers and dNTPs, and subjected to cycle sequencing reaction using BigDye Terminator v3.2 (Invitrogen Inc., USA) following vendor's protocol. The primers used for sequencing termination reaction were ITS2A, ITS2D (5'-TAT GCT TAA ATT CTG AGG GT-3'), D2A (5'-AGT CGT GTT GCT TGA TAG TGC AG-3') [34], D2B (5'-TTG GTC CGT GTT TCA AGA CGG G-3') [34], D3A (5'-GAC CCG TCT TGA AAC ACG GA-3') [33] and D3B. The sequence termination reaction products were cleaned up using ethanol precipitation and were electrophoresed in ABI Prism 3730xl. The numbers of samples sequenced from different parts of India has been shown in Table 1.

### Cloning and sequencing

For cloning, the rDNA spanning partial 5.8S, ITS2 and partial 28S was first amplified from six individual female mosquitoes, two each from Nuh (representing northern India), Gadchiroli (central India) and Chennai (representing southern India) using a high-fidelity DNA Taq polymerase to minimize PCR-errors. The PCR reaction mixture (25 μL) contained 0.5 μM of

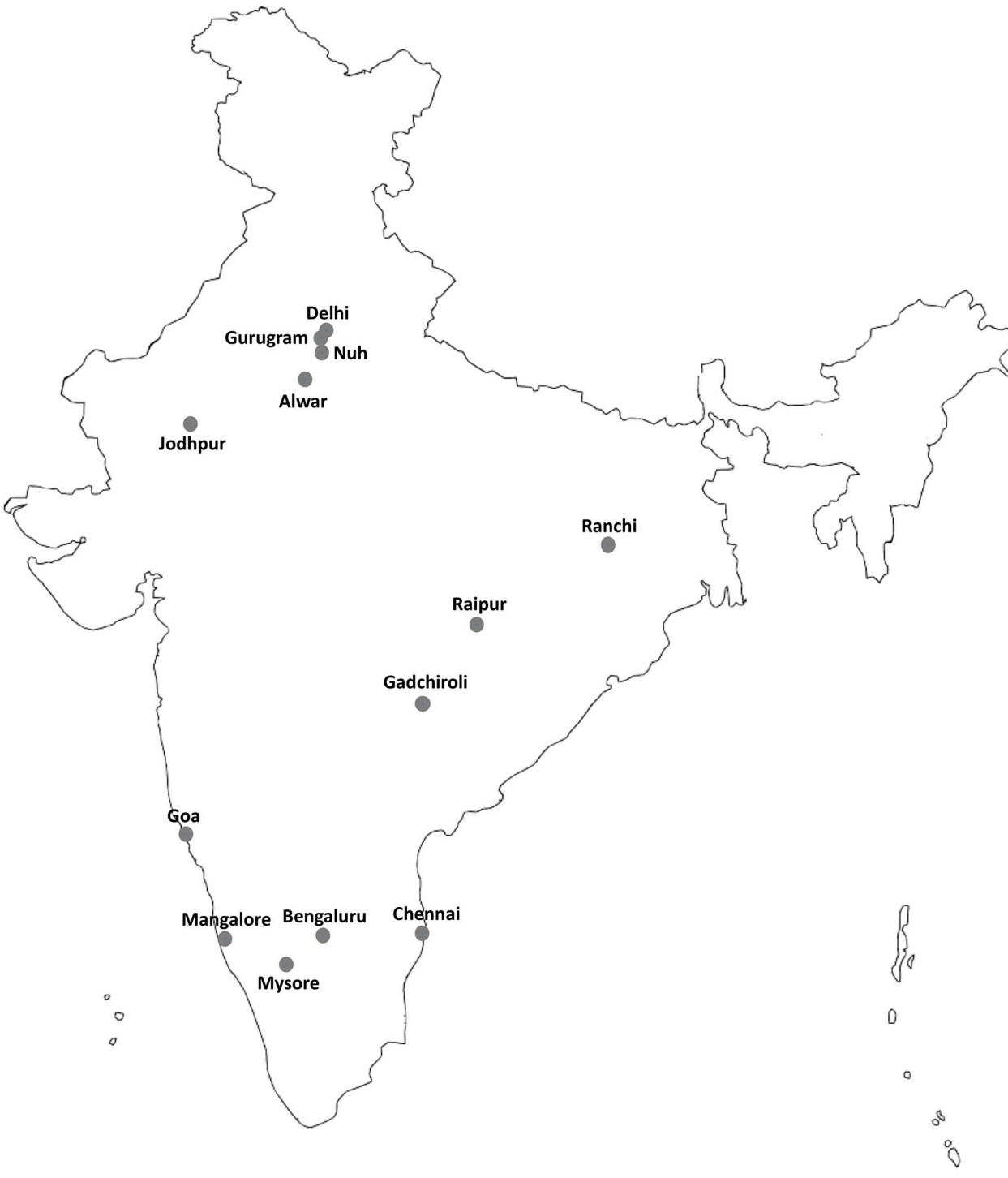

**Fig 1. Geographical location of mosquito sampling sites.**

each forward and reverse primer (ITS2A and D3B), 0.50 unit of Phusion® High-Fidelity DNA Polymerase, 5 μL of 5X Phusion HF reaction buffer (all from New England Biolabs, USA), 0.5 μL of 10 mM premixed dNTPs and 0.5 μL of gDNA. PCR conditions were: initial denaturation at 98˚C for 30 sec, 35 cycles of each of denaturation at 98˚C for 10 sec, annealing at 65 ˚C

**Table 1. Number of samples subjected to direct sequencing of rDNA regions.**

| Population | Geographical coordinates | rDNA region | |
|---|---|---|---|
| | | ITS2 | 28S (d1-d3) |
| 1. Delhi | 28.63, 77.15 | 2 | 0 |
| 2. Gurugram (Haryana) | 28.43, 76.97 | 4 | 4 |
| 3. Nuh (Haryana) | 28.10, 76.99 | 4 | 4 |
| 4. Alwar (Rajasthan) | 27.55, 76.62 | 4 | 4 |
| 5. Ranchi (Jharkhand) | 23.36, 85.32 | 9 | 9 |
| 6. Raipur (Chhatishgarh) | 21.26, 81.63 | 1 | 1 |
| 7. Gadchiroli (Maharashtra) | 20.18, 79.99 | 2 | 2 |
| 8. Goa (UT) | 15.36, 74.03 | 4 | 4 |
| 9. Bengaluru (Karnataka) | 13.01, 77.61 | 2 | 2 |
| 10. Mangalore (Karnataka) | 12.92, 74.85 | 4 | 4 |
| 11. Mysuru (Karnataka) | 12.31, 76.64 | 3 | 3 |
| 12. Chennai (Tamil Nadu) | 13.06, 80.23 | 5 | 5 |
| 13. Laboratory colony: "type form" (Jodhpur, Rajasthan) | | 2 | 2 |
| 14. Laboratory colony: "*var mysorensis*" (Jodhpur, Rajasthan) | | 2 | 2 |
| | | 48 | 46 |

for 30 sec and extension at 72 ˚C for 45 sec, followed by a cycle of final extension at 72 ˚C for 2 min. Five μL of PCR products were visualized on 1.5% agarose gel and the remaining were purified using QIAquick PCR purification kit (Qiagen Inc, USA) following the manufacturer's protocol. A-tail was incorporated at 3' end of the purified PCR product by incubating 50 ng of PCR product at 70 ˚C for 20 min in a reaction mixture (20 μL) containing 1X PCR buffer, 1.5 mM $MgCl_2$, 0.5 unit of Taq DNA polymerase, and 200 μM of dATP. Four μL of purified A-tailed PCR products were ligated to pGEM-T easy vector (Promega Corporation) and 5 μL of this ligated mixture was then transformed into DH5α competent cells (New England Biolab, USA). Transformed cultures were plated on Luria-Bertani (LB) agar plates containing 5-bromo-4-chloro-3-indolyl-beta-D-galactopyranoside (X-gal), isopropyl-beta-D-thiogalacto-pyranoside (IPTG), and 50 μg/mL ampicillin. White transformant colonies originating from each mosquito were picked up and DNA was isolated by boiling them in TE buffer at 95 ˚C. The plasmid DNA was subjected to PCR using primers ITS2 and D3B and the amplified products with successful insert, as evident from the size of PCR product on agarose gel, were treated with Exo-Sap. Sequencing of plasmid DNA was performed using primers ITS2A, ITS2D, D2A D2B, D3A and D3B.

## Results

Sequencing of cloned PCR products revealed the presence of a total of ten haplotypes. The sequences of all haplotypes are available at GenBank (accession numbers MW676288—MW676295, MW732930—MW732931). Since NCBI no more accept annotation of rDNA sequences submitted, due to a recent policy change, annotation details have been provided in S1 File. Polymorphism in nucleotide sequences were restricted to ITS2 region only and no polymorphism was recorded in flanking 5.8S and d1 to d3 domains of 28S. The alignment of ITS2 sequences in respect to all haplotypes have been shown in Fig 2. All the haplotypes resulted due to SNPs at three fixed loci in the ITS2 region, i.e., C>T transition at nucleotide base positions 172 and 383, G>A transition at 401, and two bp indel (CA) at base positions 182–183 (base position numbering as per Fig 2). To avoid inclusion of an SNP in cloned samples resulting from possible PCR error, any SNP which was present only in one clone, were

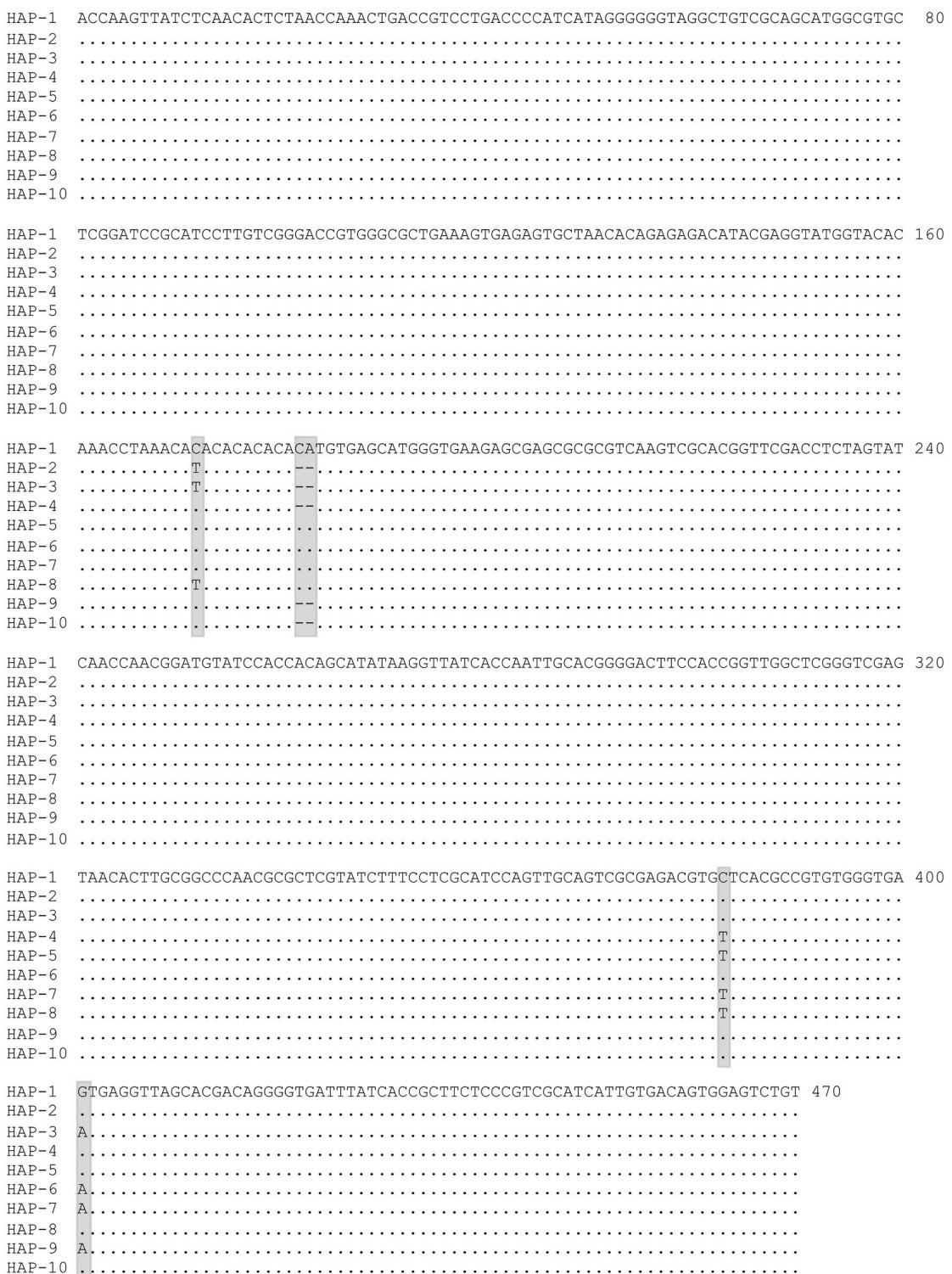

**Fig 2. Alignment of haplotypes of complete ITS2 sequence of *An. stephensi* (the flanking 5.8S and 28S rDNA sequences are not shown here).** The polymorphic loci have been highlighted (SNPs at three base positions, i.e., 172, 383 and 401; two bp indel at base positions 182–183). Dot represents similarity with Hap-1 sequence and dash represents gap in nucleotide sequence.

**Table 2. Distribution of haplotypes in clonal DNA samples (nucleotide base position numbering based on the alignment displayed in Fig 2).**

| Haplotypes | Base positions | | | | | Number of clones | | | |
|---|---|---|---|---|---|---|---|---|---|
| | 1 7 2 | 1 8 2 | 1 8 3 | 3 8 3 | 4 0 1 | Nuh (northern India) | Gadchiroli (central India) | Bengaluru (southern India) | Total |
| Hap-1 | C | C | A | C | G | 12 | 11 | 7 | 30 |
| Hap-2 | T | - | - | C | G | 4 | 2 | 3 | 9 |
| Hap-3 | T | - | - | C | A | 3 | 1 | 3 | 7 |
| Hap-4 | C | - | - | T | G | 3 | 2 | 1 | 6 |
| Hap-5 | C | C | A | T | G | 2 | 1 | 2 | 5 |
| Hap-6 | C | C | A | C | A | 1 | 0 | 2 | 3 |
| Hap-7 | C | C | A | T | A | 0 | 0 | 1 | 1 |
| Hap-8 | T | C | A | T | G | 0 | 0 | 1 | 1 |
| Hap-9 | C | - | - | C | A | 0 | 1 | 0 | 1 |
| Hap-10 | C | - | - | C | G | 0 | 1 | 0 | 1 |
| Total | | | | | | 25 | 19 | 20 | 64 |

suspected to be raised from PCR error and were excluded from the analysis. The number of clones sequenced and distribution of haplotypes in representative samples from three populations have been shown in Table 2. The Hap-1 was the dominant haplotype.

Direct sequencing of *An. stephensi* from different localities, representing northern, central, and southern India, was also performed for ITS2 and d1-d3 domains of 28S-rDNA (Table 1). Due to the presence of indel at base positions 182–183 of ITS2, all sequences were ambiguous in sequence reads beyond this point (Fig 3). Therefore, forward sequences were read before the indel point and reverse sequences read after the indel. Mixed bases were seen at all polymorphic loci, as seen in clonal sequencing, in all the samples. However, it was difficult to recognize transition 383C>T in some samples and differentiate it from the noise due to the low peak area of T-allele. The result shows that intragenomic variation is universal, at least in the Indian population. The direct sequence chromatogram also showed that CA-microsatellite present in ITS2 (base positions 172–182) was dimorphic (with six or seven repeats) and not highly polymorphic like other microsatellites. No polymorphism was seen in the coding rDNA (5.8S and d1 to d3 domains of 28S).

To investigate if mixed haplotypes in *An. stephensi* is due to the hybridization of type form and *var. mysorensis*, two individuals of each form were also sequenced for ITS2 and 28S rDNA. No difference was found in the sequence and pattern of intragenomic variations.

## Discussion

Ribosomal DNA sequence is a widely used molecular taxonomic marker for species-identification/delimitation and phylogenetic studies due to homogenization of copies of rDNA in an interbreeding population through observed concerted evolution, which leads to a remarkably low or complete absence of intra-specific variation. Among rDNA regions, the ITS2-rDNA is the most frequently used molecular marker that exhibits remarkably high inter-specific differentiation due to high evolution rate, although it remains under evolutionary constraints to maintain the specific secondary structures that provide functionality [35] of rDNA. However, the use of rDNA in molecular systematics and phylogenetic studies sometimes becomes problematic due to intragenomic variation, particularly indel. Intragenomic variations in rDNA, particularly in noncoding rDNA (ITS2 and IGS), are not uncommon and have been observed in several organisms including anophelines [36–40]. The indel in some copies of rDNA

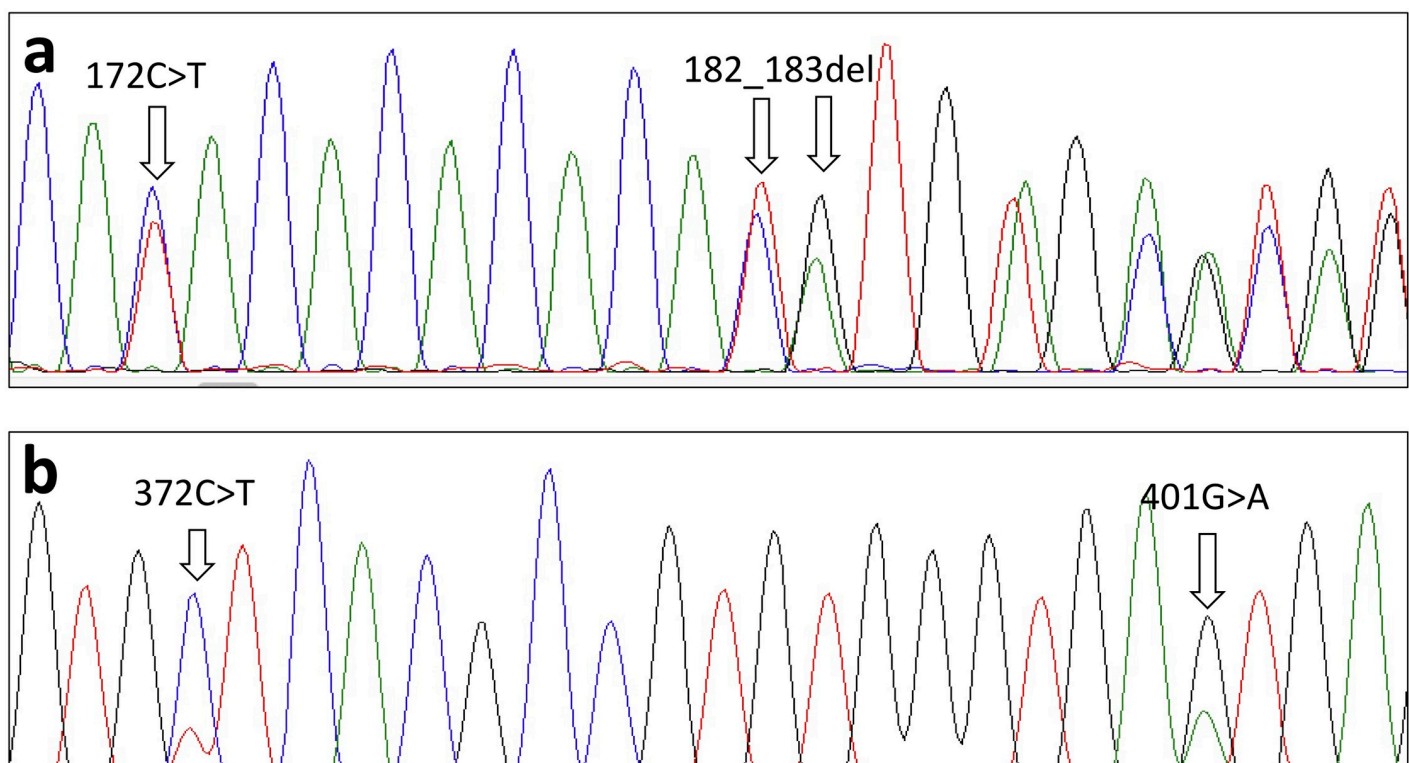

**Fig 3. Snapshots of portions of ITS2 chromatogram showing polymorphic sites.** Polymorphism 172C>T and 182_183del can be seen in forward sequence (a) and 372C>T and 401G>A can be seen in reverse sequence (b) chromatogram. The ambiguity (mixed bases) in sequences can be seen from the point of indel onwards (a).

sequence significantly affects the DNA sequence quality which leads to the collapse of the DNA sequence chromatogram [24] downstream to the point of indel. Instances exist where such intragenomic variations are overlooked and potentially inaccurate sequences are submitted to GenBank or documented in the research articles. A recent report by Sindhania et al. [25] revealed that most of the ITS2-sequences of a molecular form of *An. subpictus* (prevalent mainly in Indian mainland, Form A) available in the public domain are incorrect due to the presence of indel in one haplotype that was unnoticed by the investigators. Therefore, the generation of quality sequence and documentation of the presence of intragenomic variation in rDNA is essential for the correct molecular characterization of a species.

The ITS2 sequences of *An. stephensi* have been reported by several authors but intragenomic variations remained unnoticed. However, significant intra-specific variations in ITS2 have been reported. It is suspected that the observed intraspecific polymorphism in ITS2 is raised due to compromised sequence quality. Such intraspecific variations have been exploited for population genetic analysis [26] which may be misleading. In a study carried out in Sri Lanka [8], only a part of sequences, representing approximately half of the sequence generated (using primers from 5.8S and 28S rDNA), were submitted. Similarly, partial sequences were submitted from the Horn of Africa [2]. Trimming of significant portions of DNA sequence in these studies is probably due to ambiguities encountered in sequence read due to intragenomic variation particularly indel at the microsatellite locus. However, none of the reports documented intragenomic variation except in one GenBank entry from the Horn of Africa [2]

(accession number: MN826065) where mixed bases (G/A) have been shown at base position 401. It is most likely that intragenomic variation is present beyond India which is evident from the fact that the sequences available from the public domain from other countries are either divergent or only fragments of the sequences have been reported. However, the presence of intragenomic variation in other countries needs confirmation. This study also revealed that polymorphic sites are fixed in all Indian populations studied. A similar observation has been documented in the case of *An. subpictus* (molecular Form A) where two ITS2 haplotypes were found present in all populations from northern India to Sri Lanka [25].

The presence of intragenomic sequence variation in *An. stephensi* is indicative of less efficient concerted evolution acting on rDNA which can be due to the presence of multiple clusters of rDNA in the genome. Although the exact mechanisms of observed concerted evolution remain unclear, the homogenization of rDNA is believed to be due to a variety of genomic mechanisms of turnover such as unequal crossing over and gene conversion [41]. However, the concerted evolution is predicted to act less efficiently upon dispersed rDNA units [42] due to less frequent recombination on heterologous chromosomes [43]. Localization of rDNA clusters in *An. stephensi* through fluorescent *in-situ* hybridization (FISH) have revealed the presence of rDNA on both X and Y chromosomes [44], similar to *Drosophila melanogaster* [45] where interchromosomal exchanges are likely to be less efficient to homogenize rDNA. We noted intragenomic variations in ITS2 (non-coding rDNA) only but no evidence of intragenomic variation was found in coding rDNA sequenced in this study (partial 5.8S and d1-d3 domains of 28S-rDNA). Several reports cited by Keller et al. [46] have shown that often such intragenomic variation is restricted to the noncoding rDNA regions.

Due to limited numbers of clones sequenced, the possibility of missing some rare variants, if any, may exist; although, such variations, if present in very low frequency, will not be affecting the quality of direct sequencing chromatogram and utility of sequence in species delimitation or phylogenetic studies. It has been shown [47] that DNA copies that are present in less than one-tenth proportion do not have obvious signals in sequence chromatogram. All the variations we observed through clonal sequencing in *An. stephensi* can be seen in a direct-sequencing chromatogram. Thus intragenomic variations can also be detected to a greater extent through careful examination of direct sequencing chromatograms through the eye without the need for cloning. However, the quality of DNA sequence with minimal noise at the base of chromatogram is crucial. The noise in the DNA sequence can be significantly reduced by choosing a sequencing primer different than the primers that were used for PCR amplification. This eliminates parallel sequencing of PCR artifacts such as non-specific PCR products present in very low quantity and primer-dimers.

The second internal transcribed spacer region has been an extensively used molecular marker for species characterization, however, coding regions of rDNA have also been used for this purpose. The 28S-rDNA, which is believed to be highly conserved, has less frequently been used in molecular taxonomy. However, the variable regions of 28S-rDNA, i.e., D2 and D3 domain [18–23] have exhibited the taxonomic significance and have been used for developing molecular differentiation of closely related sibling species/groups. This study provides the first-ever reference sequence of *An. stephensi* for d1 to d3 domains of 28S rDNA. This region didn't exhibit obvious intragenomic sequence variation. Therefore, this sequence can be used as a reference sequence for *An. stephensi* for molecular taxonomy.

## Conclusions

Due to the presence of intragenomic sequence variations in the ITS2-rDNA of *An. stephensi*, this region is not a suitable molecular marker for the identification of *An. stephensi* and

molecular phylogenetic analysis. The 28S-rDNA (d1 through d3 domains) sequence can serve as a superior molecular marker for this purpose due to the absence of intragenomic variation.

## Supporting information

**S1 File. Annotation features of GenBank entries.**
(PDF)

## Acknowledgments

The authors are thankful to Dr. Robin Marwal for providing specimens of type form and *var mysorensis*, Dr. V.P. Ojha for providing mosquitoes from Gadchiroli, Mr. Shri Bhagwan for collecting mosquitoes from Chennai and Mr Uday Praksh and Mr NS Bhakuni for technical support.

## Author Contributions

**Conceptualization:** Om P. Singh.

**Data curation:** Om P. Singh.

**Formal analysis:** Om P. Singh.

**Investigation:** Om P. Singh.

**Methodology:** Shobhna Mishra, Gunjan Sharma, Om P. Singh.

**Project administration:** Om P. Singh.

**Resources:** Manoj K. Das, Om P. Singh.

**Supervision:** Om P. Singh.

**Writing – original draft:** Om P. Singh.

**Writing – review & editing:** Veena Pande.

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
