## [Decision Letter · Decision Letter 0]

14 May 2021

PONE-D-21-08154

Intragenomic sequence variations in the second internal transcribed spacer (ITS2) ribosomal DNA of the malaria vector Anopheles stephensi

PLOS ONE

Dear Dr. Singh,

Thank you for submitting your manuscript to PLOS ONE. After careful consideration, we feel that it has merit but does not fully meet PLOS ONE’s publication criteria as it currently stands. Therefore, we invite you to submit a revised version of the manuscript that addresses the points raised during the review process.

Please address all the comments from both reviewers. Discuss possible genomic reasons for the presence of intragenomic variation in rDNA. Can it be due to the presence of more than one copy of rDNA cluster in the genome of An. stephensi? For example, se the paper that shows the presence of the rDNA loci on both X and Y chromosomes, which may have different evolutionary trajectories. PMID: 25244985, PMCID: PMC4195908, DOI: 10.1186/s13059-014-0459-2

We look forward to receiving your revised manuscript.

Kind regards,

Igor V. Sharakhov

Academic Editor

PLOS ONE

Journal Requirements:

4. In your Methods section, please provide additional location information of the collection sites, including geographic coordinates for the data set if available.

5. In your Methods section, please provide additional information regarding the permits you obtained for the work. Please ensure you have included the full name of the authority that approved the collection sites access and, if no permits were required, a brief statement explaining why.

"SM and GS were supported by ICMR-Senior Research Fellowship."

 "OPS: Department of Science & Technology, India, https://dst.gov.in/

7.We note that Figure(s) 1 in your submission contain map images which may be copyrighted. All PLOS content is published under the Creative Commons Attribution License (CC BY 4.0), which means that the manuscript, images, and Supporting Information files will be freely available online, and any third party is permitted to access, download, copy, distribute, and use these materials in any way, even commercially, with proper attribution. For these reasons, we cannot publish previously copyrighted maps or satellite images created using proprietary data, such as Google software (Google Maps, Street View, and Earth). For more information, see our copyright guidelines: http://journals.plos.org/plosone/s/licenses-and-copyright.

a)  You may seek permission from the original copyright holder of Figure(s) 1 to publish the content specifically under the CC BY 4.0 license. 

Reviewers' comments:

Reviewer's Responses to Questions

**Comments to the Author**

1. Is the manuscript technically sound, and do the data support the conclusions?

Reviewer #1: Yes

Reviewer #2: Yes

2. Has the statistical analysis been performed appropriately and rigorously? 

Reviewer #1: N/A

Reviewer #2: Yes

3. Have the authors made all data underlying the findings in their manuscript fully available?

Reviewer #1: Yes

Reviewer #2: Yes

4. Is the manuscript presented in an intelligible fashion and written in standard English?

Reviewer #1: Yes

Reviewer #2: Yes

5. Review Comments to the Author

Reviewer #1: 1-The manuscript is well written; however, it is unclear whether samples collected from different locations represent mysorensis, type or intermediate forms.

2-It is mentioned in the manuscript that "28S rDNA sequence can serve as molecular markers due to absence of obvious interagenomic variations", however, absence of interagenomic variations in 28S rDNA sequences are not shown in any figures.

3-Line 159: Please correct the Accession number.

Also, correct the last two Accession numbers in Supplementary information S1 (Annotation Features of GenBank entries).

Reviewer #2: Other molecular markers such as OBP1, COI, COII has been used for molecular identification of An. stephensi. The authors must explain why ITS2 and 28S rDNA were chosen, while OBP1 is one the most appropriate choice. I suggest to reflect this in the Introduction and Discussion sections.

6. PLOS authors have the option to publish the peer review history of their article (what does this mean?). If published, this will include your full peer review and any attached files.

Reviewer #1: No

Reviewer #2: No

---

## [Author Response · Author response to Decision Letter 0]

27 May 2021

Response to Editor/Reviewers’ comments

Editor’s comment

Discuss possible genomic reasons for the presence of intragenomic variation in rDNA. Can it be due to the presence of more than one copy of rDNA cluster in the genome of An. stephensi? For example, se the paper that shows the presence of the rDNA loci on both X and Y chromosomes, which may have different evolutionary trajectories. PMID: 25244985, PMCID: PMC4195908, DOI: 10.1186/s13059-014-0459-2 

Authors’ response:

Thanks for the careful review of the manuscript and for suggesting to discuss the possible genomic reason for the presence of intragenomic variation in rDNA. We have incorporated a paragraph (lines 227-240; copied below), discussing the above. Please feel free to suggest if further edit is required for the improvement of the manuscript.

“The presence of intragenomic sequence variation in An. stephensi is indicative of less efficient concerted evolution acting on rDNA which can be due to the presence of multiple clusters of rDNA in the genome. Although the exact mechanisms of observed concerted evolution remain unclear, the homogenization of rDNA is believed to be due to a variety of genomic mechanisms of turnover such as unequal crossing over and gene conversion [41]. However, the concerted evolution is predicted to act less efficiently upon dispersed rDNA units [42] due to less frequent recombination on heterologous chromosomes [43]. Localization of rDNA clusters in An. stephensi through fluorescent in-situ hybridization (FISH) have revealed the presence of rDNA on both X and Y chromosomes [44], similar to Drosophila melanogaster [45] where interchromosomal exchanges are likely to be less efficient to homogenize rDNA. We noted intragenomic variations in ITS2 (non-coding rDNA) only but no evidence of intragenomic variation was found in coding rDNA sequenced in this study (partial 5.8S and d1-d3 domains of 28S-rDNA). Several reports cited by Keller et al. (2006) [46] have shown that often such intragenomic variation is restricted to the noncoding rDNA regions.”

Reviewer #1: 

1-The manuscript is well written; however, it is unclear whether samples collected from different locations represent mysorensis, type or intermediate forms.

Authors’ response:

Mosquito samples used in this study were collected from the remote areas and preserved on site for molecular study. To classify their variant forms, we need to transport live mosquitoes, feed them on blood meal, allow them to lay their eggs individually and count ridges on egg floats, which we could not perform due to logistic reasons.

2-It is mentioned in the manuscript that "28S rDNA sequence can serve as molecular markers due to absence of obvious interagenomic variations", however, absence of interagenomic variations in 28S rDNA sequences are not shown in any figures.

Authors’ response:

Perhaps your question is related to the Figure displaying the alignment of DNA sequences. Alignment of the entire rDNA region we sequenced (1.7 kb) requires a lot of space (can be covered in 4 pages). We, therefore, preferred to show the alignment of ITS2 only to exhibit polymorphic sites and to assign the base position numbering of each SNP/indel. These numberings were used as identifier of each SNPs/indel in the manuscript. Because no SNP was observed in coding rDNA, I think there is no significance of displaying sequence alignment of these regions. The entire sequence of all haplotypes, however, is available at GenBank. 

3-Line 159: Please correct the Accession number.

Also, correct the last two Accession numbers in Supplementary information S1 (Annotation Features of GenBank entries).

Authors’ response:

Thanks for the careful observation. We have corrected the accession numbers.

Reviewer #2: 

Other molecular markers such as OBP1, COI, COII has been used for molecular identification of An. stephensi. The authors must explain why ITS2 and 28S rDNA were chosen, while OBP1 is one the most appropriate choice. I suggest to reflect this in the Introduction and Discussion sections.

Authors’ response:

We have mentioned the use of COI-mtDNA (COI and CytB) and OBP1 for the molecular identification of An. stephensi and also mentioned the proposal of the presence of three sibling species based on the polymorphism in the intronic region of this gene (Gholizadeh et al., 2015; Firooziyan et al., 2018; lines 55-60). Regarding the explanation why ITS2 and 28S rDNA was chosen (for the study), I would like to clarify that I have not chosen rDNA ‘for the identification’ but the purpose of this communication is to report intragenomic sequence variations in rDNA of An. stephensi and associated consequences on species delimitation and phylogenetic studies. This study by no means is proposing the use of rDNA as an alternative to other markers.

---

## [Editor Report · Decision Letter 1]

31 May 2021

Intragenomic sequence variations in the second internal transcribed spacer (ITS2) ribosomal DNA of the malaria vector Anopheles stephensi

PONE-D-21-08154R1

Dear Dr. Singh,

We’re pleased to inform you that your manuscript has been judged scientifically suitable for publication and will be formally accepted for publication once it meets all outstanding technical requirements.

Kind regards,

Igor V. Sharakhov

Academic Editor

PLOS ONE
---

## [Editor Report · Acceptance letter]

4 Jun 2021

PONE-D-21-08154R1 

Intragenomic sequence variations in the second internal transcribed spacer (ITS2) ribosomal DNA of the malaria vector *Anopheles stephensi*

Dear Dr. Singh:

I'm pleased to inform you that your manuscript has been deemed suitable for publication in PLOS ONE. Congratulations! Your manuscript is now with our production department. 

Kind regards, 

on behalf of

Dr Igor V. Sharakhov 

Academic Editor

PLOS ONE